# Electro-Oxidative C3-Selenylation of Pyrido[1,2-*a*]pyrimidin-4-ones

**DOI:** 10.3390/molecules28052206

**Published:** 2023-02-27

**Authors:** Jianwei Shi, Zhichuan Wang, Xiaoxu Teng, Bing Zhang, Kai Sun, Xin Wang

**Affiliations:** 1School of Chemistry and Chemical Engineering, Yangtze Normal University, Chongqing 408100, China; 2College of Chemistry and Chemical Engineering, Yantai University, Yantai 264005, China; 3College of Chemistry, Zhengzhou University, Zhengzhou 450001, China

**Keywords:** electrochemistry, selenylation, radical, carbocation, *N*-heterocycles

## Abstract

In this work, we achieved a C3-selenylation of pyrido[1,2-*a*]pyrimidin-4-ones using an electrochemically driven external oxidant-free strategy. Various structurally diverse seleno-substituted *N*-heterocycles were obtained in moderate to excellent yields. Through radical trapping experiments, GC-MS analysis and cyclic voltammetry study, a plausible mechanism for this selenylation was proposed.

## 1. Introduction

*N*-heterocycles hold a privileged position in the preparation of drugs, agrochemicals, polymers, and other functional materials [1,2]. According to statistics, nitrogen species are presented in more than 80% of the top 200 pharmaceuticals, and two thirds of these N-containing medicines contain N-heterocyclic skeletons [3]. Among these, *N*-fused pyrido[1,2-*a*]pyrimidin-4-ones are one of the most prominent classes of structural motifs due to their ubiquity and bioactivity as the backbones of many natural and pharmacologic products [4,5,6]. A variety of derivatives based on this backbone show versatile bioactivities, including antioxidants, antipsychotics, and antiulcer drugs, etc. (Figure 1A) [7,8,9,10]. During the past decades, many efforts have been devoted to the construction and derivatization of such *N*-fused heterocycles, mainly including multicomponent cyclization, metal catalyzed direct C−H functionalization and metal-free chalcogenation with extra stoichiometric oxidants [11,12,13,14,15,16]. However, inevitable metal residue, extra stoichiometric oxidants, harmful halogenated solvents and inert gas conditions seriously restrict use for pharmaceutical chemistry applications. Thus, the development of modular approaches that provide facile and practical access to functionalized pyrido[1,2-*a*]pyrimidin-4-ones continues to be in high demand.

Selenium-containing compounds play important roles in organic synthesis, medicinal chemistry, and biochemistry [17,18,19,20,21]. In particular, researchers have demonstrated that *N*-heterocycles modified with organylselanyl groups exhibit unique pharmacological activities and physicochemical properties and thereby have higher applied value (Figure 1B). In the long history of selenium chemistry, diselenides as readily available substrates [22,23,24,25,26,27,28] or precatalysts [29,30,31,32,33,34] have garnered considerable attention for use in various reactions. Especially in the last five years, electrochemistry-induced C-H bond selenylation for the synthesis seleno-heterocycles has been booming [35,36,37,38,39,40,41,42,43,44]. Although selenium can bring positive physiochemical properties of bioactive molecules and drugs, the methods for direct selenylation of pyrido[1,2-*a*]pyrimidin-4-ones are still limited. Until 2021, the only two examples for C-3 selenylation of pyrido[1,2-*a*]pyrimidin-4-ones by Das group was established (Figure 1A) [45,46]. These achievements may be important; however, practical applications of the above-mentioned synthetic strategies are limited to the stoichiometric or excessive oxidants, diselenides, harmful halogenated solvents and the difficult collection of the target products from large amounts of unexpected byproducts and unconsumed reagents. Electrochemical technology employe traceless electrons as redox reagents, avoiding extra chemical oxidants, reductants, and transition-metal catalysts, and more importantly, it bears the unique advantage of controlling reactivity by “dialing-in” the specific potential on demand [47,48,49,50,51,52,53,54]. We envisioned whether a more easy-going radical selenylation of the pyrido[1,2-*a*]pyrimidones via electrochemical technology may be realized, which would afford a sustainable and universal selenylation method (Figure 1B).

## 2. Results and Discussion

In order to optimize the reaction conditions for the anticipated selenylation of pyrido[1,2-*a*]pyrimidin-4-ones, we commenced our study by employing 2-phenyl-4*H*-pyrido[1,2-*a*]pyrimidin-4-one **1a** and diphenyl diselenide **2a** as model substrates in this reaction. As shown in Table 1, Pt(+)/Pt(−) were chosen as both the anode and cathode, *^n^*Bu_4_NBF_4_ as the supporting electrolyte, reactions were performed in MeCN at 60 °C under 5V constant voltage in an undivided three-necked bottle, for 3 h, and the target **3a** could be isolated in 42% isolated yield (entry 1). Other electrolytes commonly used for electrochemical conditions such as *^n^*Bu_4_NI, *^n^*Bu_4_NPF_6_ and *^n^*Bu_4_NClO_4_ were then tested. The results showed that *^n^*Bu_4_NPF_6_ exhibited a positive effect, leading to the isolated **3a** with a satisfactory 66% yield, while *^n^*Bu_4_NI and *^n^*Bu_4_NClO_4_ did not proceed efficiently (entries 2−4). Further solvent screening revealed that DMF, DMSO, MeOH and HFIP are not ideal options for this transformation (entries 5−8). Moreover, the effects of the electrode materials were explored. However, lower reaction yields were obtained when the Pt(+)/Pt(−) was replaced by C(+)/C(−) and C(+)/Pt(−) (entries 9 and 10). When the reaction temperature was adjusted from 60 to 40 °C or to room temperature, the yields dramatically decreased (entries 11 and 12). When the reaction time is extended to 5 h, the yield of **3a** can be increased sharply to 94% (entry 13). The control experiment also showed that no desired product 3a was generated without electricity (entry 14).

With the optimized conditions in hand, we further evaluated the scope of the substrates by examining various functionalized pyrido[1,2-*a*]pyrimidin-4-ones **1**, and the results are illustrated in Table 2. As can be seen, for substrates bearing 2-Me, 3-Me, 3-Cl and 4-OMe on the pyridine ring, this transformation could be proceeded smoothly to provide the corresponding **3b**−**3e** in 67−96% yields. Furthermore, 7-phenyl-5*H*-thiazolo[3,2-*a*]pyrimidin-5-one **1f** was compatible with this conversion, giving the corresponding product **3f** in 82% yield. Substituents at the 7-position can also vary from aryl to methyl, with the desired products **3g**−**3j** isolated in 67−96% yields. In further demonstration of the utility and applicability of this method, a gram-scale selenylation reaction with **1a** was performed. The gram-scale reaction proceeded well to form the corresponding product **3a** in 91% yield, demonstrating the capacity to apply the protocol.

We next focused our attention toward evaluating the scope of various diselenides (Table 3). Regardless of electron-donating (2-OMe, 3-Me, 4-Me, 4-OMe,) or electron-withdrawing groups (2-CF_3_, 3-Br, 4-Cl, 4-Br) on the phenyl ring of the selenide moiety, this electro-oxidative C3-selenylation could proceed smoothly, giving the corresponding products **3k**−**3r** in moderate to excellent yields (60–97%). Multi-substituted diselenides, 1,2-di(naphthalen-2-yl)diselane, 1,2-di(pyridin-2-yl)diselane and 1,2-dimethyldiselane were also compatible with this transformation, producing the corresponding products **3s**−**3y** in moderate to excellent yields (40–97%). Possibly due to the strong oxidation environment, the selenylation yields with the electron-rich diaryl diselenides were significantly lower (**3t** and **3u**). The electronic and steric effects with diselenides have no obvious effects on the reaction. When substituents at the 7-position varied from aryl to methyl, the electro-oxidative C3-selenylation with 3-Br, 3-Me, 4-Me and 4-Cl substituted diselenides and 1,2-dimethyldiselane proceeded smoothly, delivering the desired products **3aa**−**3ad** in 73−95% yields. Meanwhile, 7-methyl-5*H*-thiazolo[3,2-*a*]pyrimidin-5-one was also a good partner in this transformation, and selenylated **3ae** could be isolated in 85% yield.

Mechanistic information was collected to elucidate the detailed reaction pathways. First, radical trapping experiments were performed. When 2 equiv of TEMPO (2,2,6,6-tetramethyl-1-piperidinyloxy) or BHT (2,4-di-*tert*-butyl-4-methylphenol) was added into the reaction system, the desired product **3a** was totally suppressed. Furthermore, adduct **4** was observed through GC-MS analysis (Figure 2a,b). When 2 equiv of stilbene was added, adducts **5** and **6** were observed through GC-MS analysis (Figure 2c). These results indicated that this reaction mostly proceeds via a radical pathway.

Second, the cyclic voltammetry (CV) experiments on both reactants were carried out. The measured oxidation peak of **1a** presented at 1.98 V (Figure 2, blue line), and an obvious oxidation peak of diphenyl diselenide **2a** could be observed at 1.88 V (Figure 2, red line). Since the reactions were performed under 5V constant voltage, both **1a** and **2a** may undergo single-electron oxidation, and the radical trapping experiments also demonstrated this result (Figure 2b,c).

On the basis of mechanistic studies and previous literature reports [45,46,55,56,57], the proposed mechanism of electro-oxidative C3-selenylation of pyrido[1,2-*a*]pyrimidin-4-ones is depicted in Figure 3. Firstly, the anodic oxidation of diselenide **2a** could deliver PhSe^.^ and PhSe^+^. Secondly, the addition of RSe^.^ on the C-3 position of 2-phenyl-4*H*-pyrido[1,2-*a*] pyrimidin-4-one **1a** generates the radical intermediate **A**. Anodic oxidation of **A** and the subsequent deprotonation results in the final products **3a**. At the cathode, protons and PhSe^+^ are reduced to H_2_ and PhSe^.^ at the surface of the cathode to complete this conversion.

However, according to radical trapping experiments, the other pathway involved the anodic oxidation of both **1a** and **2a**, which cannot be ruled out. The cross-coupling of the corresponding PhSe^.^ and carbon-centered radicals could also quickly deliver the final products **3a**.

## 3. Materials and Methods

### 3.1. Materials and Instruments

All reagents were purchased from commercial sources and used without further purification. ^1^H NMR, ^13^C NMR spectra were recorded on a Bruker Ascend™ 400 or Bruker Ascend™ 500 spectrometer (Billerica, MA, USA) in deuterated solvents containing TMS as an internal reference standard. All high-resolution mass spectra (HRMS) were measured on a mass spectrometer by using electrospray ionization orthogonal acceleration time-of-flight (ESI-OA-TOF), and the purity of all samples used for HRMS (>95%) was confirmed by ^1^H NMR and ^13^C NMR spectroscopic analysis. Melting points were measured on a melting point apparatus equipped with a thermometer and were uncorrected. All the reactions were monitored by thin-layer chromatography (TLC) using GF254 silica gel-coated TLC plates. Purification by flash column chromatography was performed over SiO_2_ (silica gel 200−300 mesh).

### 3.2. General Procedure for the Synthesis of **1**

A mixture of 2-aminopyridines (3.00 mmol) and the appropriate *β*-keto esters (4.50 mmol) in PPA (6.00 g) was heated at 100 °C for 1 h while stirring with a glass stick. The thick syrup thus obtained was slowly poured into crushed ice, and the resulting suspension was neutralized with 10% aqueous sodium hydroxide. The solid precipitate was collected by filtration, washed with water, and recrystallized to give **1** (Figure 4).

### 3.3. The General Procedure for the Synthesis of **3**

Various 2-(aryl/alkyl) substituted 4*H*-Pyrido-[1,2-*a*]-Pyrimidin-4-ones **1** (0.20 mmol), diselenide **2** (0.20 mmol), *^n^*Bu_4_NPF_6_ (0.20 mmol) and MeCN (5.0 mL) were placed in a 10 mL two-necked round-bottomed flask. The flask was equipped with a stir bar, a platinum plate (1 cm × 1 cm) anode and a platinum plate (1 cm × 1 cm) cathode. The electrolysis was carried out under air atmosphere at 60 °C using a constant potential of 5 V until complete consumption of the substrate **1** (monitored by TLC, about 5 h). After the completion of the reaction, the mixture was quenched by NaHCO_3_ (sat. aq. 150 mL) and extracted with CH_2_Cl_2_ (50 mL × 3). Then, the organic solvent was concentrated in vacuo. The residue was purified by flash column chromatography with ethyl acetate and petroleum ether as eluent to give **3**.

***2-Phenyl-3-(phenylselanyl)-4H-pyrido[1,2-a]pyrimidin-4-one* (3a)**. 2-Phenyl-8,9-dihydro-4*H*-pyrido[1,2-*a*]pyrimidin-4-one (0.20 mmol, 44.42 mg) was reacted with PhSeSePh (0.20 mmol, 62.43 mg) according to General Procedure. The crude product was purified by column chromatography (petroleum ether: ethyl acetate = 5:1) to afford the title compound as a yellow solid (m. p. 129–130 °C) in 94% yield (71.11 mg). **R*_f_*** (petroleum ether/ethyl acetate = 5:2): 0.24; **^1^H NMR** (500 MHz, CDCl_3_) *δ* 9.08 (d, *J* = 7.1 Hz, 1H), 7.80–7.76 (m, 1H), 7.73 (d, *J* = 8.8 Hz, 1H), 7.60 (dd, *J* = 6.5, 3.1 Hz, 2H), 7.43–7.39 (m, 3H), 7.32–7.28 (m, 2H), 7.18 (td, *J* = 7.1, 1.4 Hz, 1H), 7.15 (dd, *J* = 6.3, 2.7 Hz, 3H); **^13^C NMR** (125 MHz, CDCl_3_) *δ* 168.10, 157.78, 150.25, 140.24, 136.87, 131.83, 131.08, 129.33, 128.99, 128.89, 128.00, 127.89, 126.68, 126.64, 116.08, 105.70; **HRMS** (ESI) calcd for C_20_H_15_N_2_OSe [M+H]^+^: 379.0344, found: 379.0338.

***6-Methyl-2-phenyl-3-(phenylselanyl)-4H-pyrido[1,2-a]pyrimidin-4-one* (3b)**. 6-Methyl-2-phenyl-4*H*-pyrido[1,2-*a*]pyrimidin-4-one (0.20 mmol, 47.25 mg) was reacted with PhSeSePh (0.20 mmol, 62.43 mg) according to General Procedure. The crude product was purified by column chromatography (petroleum ether: ethyl acetate = 5:1) to afford the title compound as a yellow solid (m. p. 171–172 °C) in 96% yield (74.90 mg). **R*_f_*** (petroleum ether/ethyl acetate = 5:2): 0.45; **^1^H NMR** (500 MHz, CDCl_3_) *δ* 7.59 (dd, *J* = 6.5, 2.9 Hz, 2H), 7.51–7.46 (m, 2H), 7.41–7.35 (m, 3H), 7.28 (dd, *J* = 6.5, 2.9 Hz, 2H), 7.16–7.09 (m, 3H), 6.76–6.70 (m, 1H), 2.99 (s, 3H); ^**13**^**C NMR** (125 MHz, CDCl_3_) *δ* 166.91, 161.42, 152.72, 144.17, 139.86, 135.92, 132.26, 130.49, 129.28, 128.97, 128.90, 127.82, 126.39, 125.35, 118.85, 107.00, 24.54; **HRMS** (ESI) calcd for C_21_H_17_N_2_OSe [M+H]^+^: 393.0501, found: 393.0494.

***7-Chloro-2-phenyl-3-(phenylselanyl)-4H-pyrido[1,2-a]pyrimidin-4-one* (3c)**. 7-Chloro-2-phenyl-4*H*-pyrido[1,2-*a*]pyrimidin-4-one (0.20 mmol, 51.26 mg) was reacted with PhSeSePh (0.20 mmol, 62.43 mg) according to General Procedure. The crude product was purified by column chromatography (petroleum ether: ethyl acetate = 5:1) to afford the title compound as a yellow solid (m. p. 169–170 °C) in 70% yield (59.32 mg). **R*_f_*** (petroleum ether/ethyl acetate = 5:2): 0.56; **^1^H NMR** (500 MHz, CDCl_3_) *δ* 9.07 (d, *J* = 1.0 Hz, 1H), 7.70–7.65 (m, 2H), 7.59 (dd, *J* = 6.7, 2.6 Hz, 2H), 7.42 (dd, *J* = 5.1, 1.5 Hz, 3H), 7.31 (dd, *J* = 6.5, 2.9 Hz, 2H), 7.17–7.13 (m, 3H); **^13^C NMR** (125 MHz, CDCl_3_) *δ* 167.52, 156.78, 148.46, 139.86, 137.89, 131.52, 131.34, 129.55, 129.01, 128.94, 127.96, 127.61, 126.97, 125.64, 124.64, 107.00; **HRMS** (ESI) calcd for C_20_H_14_ClN_2_OSe [M+H]^+^: 412.9954, found: 412.9948.

***7-Methyl-2-phenyl-3-(phenylselanyl)-4H-pyrido [1,2-a]pyrimidin-4-one* (3d)**. 7-Methyl-2-phenyl-4*H*-pyrido[1,2-*a*]pyrimidin-4-one (0.20 mmol, 47.25 mg) was reacted with PhSeSePh (0.20 mmol, 62.43 mg) according to General Procedure. The crude product was purified by column chromatography (petroleum ether: ethyl acetate = 5:1) to afford the title compound as a yellow solid (m. p. 190–191 °C) in 95% yield (74.49 mg). **R*_f_*** (petroleum ether/ethyl acetate = 5:1): 0.14; ^**1**^**H NMR** (500 MHz, CDCl_3_) *δ* 8.90 (s, 1H), 7.66 (s, 2H), 7.58 (dd, *J* = 6.5, 2.8 Hz, 2H), 7.43–7.39 (m, 3H), 7.30 (dd, *J* = 6.4, 2.8 Hz, 2H), 7.16–7.12 (m, 3H), 2.44 (s, 3H); ^**13**^**C NMR** (125 MHz, CDCl_3_) *δ* 167.74, 157.65, 149.19, 140.34, 139.81, 131.98, 131.00, 129.23, 128.95, 128.87, 127.87, 126.60, 126.49, 126.08, 125.46, 105.30, 18.43; **HRMS** (ESI) calcd for C_21_H_17_N_2_OSe [M+H]^+^: 393.0501, found: 393.0495.

***8-Methoxy-2-phenyl-3-(phenylselanyl)-4H-pyrido[1,2-a]pyrimidin-4-one* (3e)**. 8-Methoxy-2-phenyl-4*H*-pyrido[1,2-*a*]pyrimidin-4-one (0.20 mmol, 50.45 mg) was reacted with PhSeSePh (0.20 mmol, 62.43 mg) according to General Procedure. The crude product was purified by column chromatography (petroleum ether: ethyl acetate = 5:1) to afford the title compound as a yellow solid (m. p. 179–180 °C) in 67% yield (54.82 mg). **R*_f_*** (petroleum ether/ethyl acetate = 5:2): 0.31; **^1^H NMR** (500 MHz, CDCl_3_) *δ* 8.58 (d, *J* = 2.6 Hz, 1H), 7.68 (d, *J* = 9.6 Hz, 1H), 7.61–7.54 (m, 3H), 7.43–7.39 (m, 3H), 7.31 (dd, *J* = 6.4, 3.0 Hz, 2H), 7.17–7.13 (m, 3H), 3.93 (s, 3H); ^**13**^**C NMR** (125 MHz, CDCl_3_) *δ* 166.72, 157.57, 151.26, 147.29, 140.26, 132.12, 131.94, 131.02, 129.21, 128.98, 128.90, 127.89, 127.34, 126.63, 107.49, 105.10, 56.57; **HRMS** (ESI) calcd for C_21_H_17_N_2_O_2_Se [M+H]^+^:409.0450, found: 409.0444.

***7-Phenyl-6-(phenylselanyl)-5H-thiazolo[3,2-a]pyrimidin-5-one* (3f)**. 7-Phenyl-5*H*-thiazolo[3,2-*a*]pyrimidin-5-one (0.20 mmol, 45.65 mg) was reacted with PhSeSePh (0.20 mmol, 62.43 mg) according to General Procedure. The crude product was purified by column chromatography (petroleum ether: ethyl acetate = 5:1) to afford the title compound as a yellow solid (m. p. 161–162 °C) in 82% yield (62.71 mg). **R*_f_*** (petroleum ether/ethyl acetate = 5:2): 0.24; **^1^H NMR** (400 MHz, CDCl_3_) *δ* 7.99 (d, *J* = 4.9 Hz, 1H), 7.59–7.54 (m, 2H), 7.43–7.39 (m, 3H), 7.32 (dd, *J* = 6.5, 3.0 Hz, 2H), 7.18–7.14 (m, 3H), 7.01 (d, *J* = 4.9 Hz, 1H); ^**13**^**C NMR** (100 MHz, CDCl_3_) *δ* 1166.97, 162.25, 158.15, 139.56, 131.54, 131.25, 129.51, 129.04, 128.97, 127.87, 126.86, 122.68, 112.29, 107.05; **HRMS** (ESI) calcd for C_18_H_13_N_2_OSSe [M+H]^+^: 384.9908, found: 384.9902.

***2-(4-Methoxyphenyl)-3-(phenylselanyl)-4H-pyrido[1,2-a]pyrimidin-4-one* (3g)**. 2-(4-Methoxyphenyl)-4*H*-pyrido[1,2-*a*]pyrimidin-4-one (0.20 mmol, 50.45 mg) was reacted with PhSeSePh (0.20 mmol, 62.43 mg) according to General Procedure. The crude product was purified by column chromatography (petroleum ether: ethyl acetate = 5:1) to afford the title compound as a yellow solid (m. p. 161–162 °C) in 67% yield (54.75 mg). **R*_f_*** (petroleum ether/ethyl acetate = 5:2): 0.18; **^1^H NMR** (500 MHz, CDCl_3_) *δ* 9.05 (d, *J* = 7.1 Hz, 1H), 7.79–7.74 (m, 1H), 7.71 (d, *J* = 8.7 Hz, 1H), 7.64 (d, *J* = 8.6 Hz, 2H), 7.31 (dd, *J* = 6.6, 2.7 Hz, 2H), 7.15 (dd, *J* = 6.7, 3.9 Hz, 4H), 6.93 (d, *J* = 8.6 Hz, 2H), 3.85 (s, 3H); **^13^C NMR** (126 MHz, CDCl_3_) *δ* 167.48, 160.70, 157.86, 150.13, 136.74, 132.59, 132.05, 130.86, 130.75, 129.00, 127.99, 126.58, 126.55, 115.81, 113.25, 104.92, 55.39; **HRMS** (ESI) calcd for C_21_H_17_N_2_O_2_Se [M+H]^+^: 409.0450, found: 409.0444.

***2-(3-Fluorophenyl)-3-(phenylselanyl)-4H-pyrido[1,2-a]pyrimidin-4-one* (3h)**. 2-(3-Fluorophenyl)-4*H*-pyrido[1,2-*a*]pyrimidin-4-one (0.20 mmol, 48.05 mg) was reacted with PhSeSePh (0.20 mmol, 62.43 mg) according to General Procedure. The crude product was purified by column chromatography (petroleum ether: ethyl acetate = 5:1) to afford the title compound as a yellow solid (m. p. 133–134 °C) in 96% yield (54.75 mg). **R*_f_*** (petroleum ether/ethyl acetate = 5:1): 0.11; **^1^H NMR** (500 MHz, CDCl_3_) *δ* 9.08 (d, *J* = 7.1 Hz, 1H), 7.83–7.76 (m, 1H), 7.72 (d, *J* = 8.8 Hz, 1H), 7.35 (t, *J* = 5.4 Hz, 2H), 7.30 (dd, *J* = 8.6, 5.5 Hz, 3H), 7.19 (t, *J* = 6.9 Hz, 1H), 7.16–7.06 (m, 4H); ^**13**^**C NMR** (126 MHz, CDCl_3_) *δ* 166.44 (d, *J* = 2.4 Hz), 163.46, 161.01, 157.76, 150.27, 142.23, 142.15, 137.12, 131.53, 131.33, 129.49 (d, *J* = 8.3 Hz), 129.06, 128.00, 126.75 (d, *J* = 26.9 Hz), 124.77 (d, *J* = 3.0 Hz), 116.30 (d, *J* = 9.8 Hz), 116.08 (d, *J* = 7.2 Hz), 115.89, 105.97; **^19^F NMR** (471 MHz, CDCl_3_) *δ* −113.07; **HRMS** (ESI) calcd for C_20_H_14_FN_2_OSe [M+H]^+^: 397.0250, found: 397.0244.

***2-Methyl-3-(phenylselanyl)-4H-pyrido[1,2-a]pyrimidin-4-one* (3i)**. 2-Methyl-4*H*-pyrido[1,2-*a*]pyrimidin-4-one (0.20 mmol, 32.04 mg) was reacted with PhSeSePh (0.20 mmol, 74.91 mg) according to General Procedure. The crude product was purified by column chromatography (petroleum ether: ethyl acetate = 5:1) to afford the title compound as a yellow solid (m. p. 136–137 °C) in 78% yield (49.24 mg). **R*_f_*** (petroleum ether/ethyl acetate = 5:2): 0.24; **^1^H NMR** (400 MHz, CDCl_3_) *δ* 9.04 (d, *J* = 7.1 Hz, 1H), 7.79–7.75 (m, 1H), 7.61 (d, *J* = 8.9 Hz, 1H), 7.41–7.35 (m, 2H), 7.22–7.12 (m, 4H), 2.73 (s, 3H); ^**13**^**C NMR** (126 MHz, CDCl_3_) *δ* 169.50, 157.25, 150.25, 136.98, 131.48, 130.58, 129.20, 128.21, 126.62, 125.86, 115.74, 105.71, 26.82; **HRMS** (ESI) calcd for C_15_H_13_N_2_OSe [M+H]^+^: 317.0188, found: 317.0182.

***7-Methyl-6-(phenylselanyl)-5H-thiazolo[3,2-a]pyrimidin-5-one* (3j)**. 7-Methyl-5*H*-thiazolo[3,2-*a*]pyrimidin-5-one (0.20 mmol, 33.24 mg) was reacted with PhSeSePh (0.20 mmol, 62.43 mg) according to General Procedure. The crude product was purified by column chromatography (petroleum ether: ethyl acetate = 5:1) to afford the title compound as a yellow solid (m. p. 187–188 °C) in 77% yield (49.67 mg). **R*_f_*** (petroleum ether/ethyl acetate = 5:2): 0.21; **^1^H NMR** (400 MHz, CDCl_3_) *δ* 7.99 (d, *J* = 4.9 Hz, 1H), 7.59–7.54 (m, 2H), 7.43–7.39 (m, 3H), 7.32 (dd, *J* = 6.5, 3.0 Hz, 2H), 7.18–7.14 (m, 3H), 7.01 (d, *J* = 4.9 Hz, 1H); ^**13**^**C NMR** (100 MHz, CDCl_3_) *δ* 168.54, 162.33, 157.74, 131.11, 130.84, 129.21, 126.79, 122.88, 111.52, 107.06, 26.41; **HRMS (ESI)** calcd for C_13_H_11_N_2_OSSe [M+H]^+^: 322.9752, found: 322.9745.

***3-((2-Methoxyphenyl)selanyl)-2-phenyl-4H-pyrido[1,2-a]pyrimidin-4-one* (3k)**. 2-Phenyl-8,9-dihydro-4*H*-pyrido[1,2-*a*]pyrimidin-4-one (0.20 mmol, 44.42 mg) was reacted with 1,2-bis(2-methoxyphenyl)diselane (0.20 mmol, 74.44 mg) according to General Procedure. The crude product was purified by column chromatography (petroleum ether: ethyl acetate = 5:1) to afford the title compound as a yellow solid (m. p. 156–157 °C) in 87% yield (70.96 mg). **R*_f_*** (petroleum ether/ethyl acetate = 5:2): 0.16; **^1^H NMR** (500 MHz, CDCl_3_) *δ* 9.08 (d, *J* = 7.1 Hz, 1H), 7.81–7.73 (m, 2H), 7.63 (dd, *J* = 6.5, 3.0 Hz, 2H), 7.37 (dd, *J* = 7.0, 3.7 Hz, 3H), 7.20–7.08 (m, 2H), 6.95 (dd, *J* = 7.8, 1.3 Hz, 1H), 6.78–6.73 (m, 2H), 3.79 (s, 3H); ^**13**^**C NMR** (125 MHz, CDCl_3_) *δ* 169.01, 157.79, 156.62, 150.46, 140.22, 137.01, 129.35, 128.84, 128.73, 128.03, 127.84, 127.07, 126.64, 121.50, 121.24, 116.08, 110.45, 103.08, 55.73; **HRMS** (ESI) calcd for C_21_H_17_N_2_O_2_Se [M+H]^+^: 409.0450, found: 409.0443.

***2-Phenyl-3-((2-(trifluoromethyl)phenyl)selanyl)-4H-pyrido[1,2-a]pyrimidin-4-one* (3l)**. 2-Phenyl-8,9-dihydro-4*H*-pyrido[1,2-*a*]pyrimidin-4-one (0.20 mmol, 44.42 mg) was reacted with 1,2-bis(2-(trifluoromethyl)phenyl)diselane (0.20 mmol, 89.6 mg) according to General Procedure. The crude product was purified by column chromatography (petroleum ether: ethyl acetate = 5:1) to afford the title compound as a yellow solid (m. p. 151–152 °C) in 65% yield (57.71 mg). **R*_f_*** (petroleum ether/ethyl acetate = 5:2): 0.14; **^1^H NMR** (500 MHz, CDCl_3_) *δ* 9.07 (dd, *J* = 7.1, 0.6 Hz, 1H), 7.86–7.81 (m, 1H), 7.78 (d, *J* = 8.4 Hz, 1H), 7.62–7.57 (m, 3H), 7.42–7.37 (m, 3H), 7.26–7.19 (m, 4H); **^13^C NMR** (125 MHz, CDCl_3_) *δ* 168.78, 157.69, 150.60, 139.74, 137.39, 131.93 (d, *J* = 6.9 Hz), 131.02, 129.62, 129.28, 129.03, 128.80, 127.98 (d, *J* = 6.8 Hz), 126.89 (q, *J* = 5.4 Hz), 126.74, 125.96, 125.15, 122.97, 116.39, 104.13 (d, *J* = 2.8 Hz); **^19^F NMR** (471 MHz, CDCl_3_) *δ* −61.18; **HRMS** (ESI) calcd for C_21_H_14_F_3_N_2_OSe [M+H]^+^: 447.0218, found: 447.0212.

***2-Phenyl-3-(m-tolylselanyl)-4H-pyrido[1,2-a]pyrimidin-4-one* (3m)**. 2-Phenyl-8,9-dihydro-4*H*-pyrido[1,2-*a*]pyrimidin-4-one (0.20 mmol, 44.42 mg) was reacted with 1,2-di-*m*-tolyldiselane (0.20 mmol, 68.04 mg) according to General Procedure. The crude product was purified by column chromatography (petroleum ether: ethyl acetate = 5:1) to afford the title compound as a yellow solid (m. p. 126–127 °C) in 60% yield (49.21 mg). **R*_f_*** (petroleum ether/ethyl acetate = 5:1): 0.11; **^1^H NMR** (500 MHz, CDCl_3_) *δ* 9.08 (d, *J* = 7.1 Hz, 1H), 7.80–7.72 (m, 2H), 7.60 (dd, *J* = 6.4, 2.9 Hz, 2H), 7.44–7.38 (m, 3H), 7.20–7.15 (m, 1H), 7.13–7.07 (m, 2H), 7.03 (t, *J* = 7.6 Hz, 1H), 6.95 (d, *J* = 7.4 Hz, 1H), 2.23 (s, 3H); ^**13**^**C NMR** (125 MHz, CDCl_3_) *δ* 167.96, 157.80, 150.21, 140.24, 138.67, 136.79, 131.78, 131.52, 129.26, 128.89, 128.78, 128.13, 128.01, 127.86, 127.63, 126.63, 116.03, 105.88, 21.33; **HRMS** (ESI) calcd for C_21_H_17_N_2_OSe [M+H]^+^: 393.0501, found: 393.0496.

***3-((3-Bromophenyl)selanyl)-2-phenyl-4H-pyrido[1,2-a]pyrimidin-4-one* (3n)**. 2-Phenyl-8,9-dihydro-4*H*-pyrido[1,2-*a*]pyrimidin-4-one (0.20 mmol, 44.42 mg) was reacted with 1,2-bis(3-bromophenyl)diselane (0.20 mmol, 93.99 mg) according to General Procedure. The crude product was purified by column chromatography (petroleum ether: ethyl acetate = 5:1) to afford the title compound as a yellow solid (m. p. 96–97 °C) in 90% yield (81.84 mg). **R*_f_*** (petroleum ether/ethyl acetate = 5:1): 0.11; **^1^H NMR** (500 MHz, CDCl_3_) *δ* 9.08 (d, *J* = 6.9 Hz, 1H), 7.84–7.79 (m, 1H), 7.75 (d, *J* = 8.9 Hz, 1H), 7.61–7.54 (m, 2H), 7.46–7.35 (m, 4H), 7.25–7.19 (m, 3H), 7.00 (t, *J* = 7.9 Hz, 1H); ^**13**^**C NMR** (125 MHz, CDCl_3_) *δ* 168.16, 157.64, 150.37, 139.97, 137.16, 133.80, 133.32, 130.27, 129.72, 129.48, 129.06, 128.79, 128.02, 127.97, 126.71, 122.83, 116.31, 105.02; **HRMS** (ESI) calcd for C_20_H_14_BrN_2_OSe [M+H]^+^: 456.9449, found: 456.9439.

***2-Phenyl-3-(p-tolylselanyl)-4H-pyrido[1,2-a]pyrimidin-4-one* (3o)**. 2-Phenyl-8,9-dihydro-4*H*-pyrido[1,2-*a*]pyrimidin-4-one (0.20 mmol, 44.42 mg) was reacted with 1,2-di-*p*-tolyldiselane (0.20 mmol, 68.04 mg) according to General Procedure. The crude product was purified by column chromatography (petroleum ether: ethyl acetate = 5:1) to afford the title compound as a yellow solid (m. p. 157–158 °C) in 94% yield (73.34 mg). **R*_f_*** (petroleum ether/ethyl acetate = 5:2): 0.27; **^1^H NMR** (500 MHz, CDCl_3_) *δ* 9.06 (d, *J* = 7.1 Hz, 1H), 7.78–7.70 (m, 2H), 7.60 (dd, *J* = 6.4, 2.8 Hz, 2H), 7.45–7.40 (m, 3H), 7.23 (d, *J* = 8.0 Hz, 2H), 7.15 (s, 1H), 6.97 (d, *J* = 7.9 Hz, 2H), 2.25 (s, 3H); ^**13**^**C NMR** (126 MHz, CDCl_3_) *δ* 167.82, 157.74, 150.13, 140.32, 136.74, 136.71, 131.64, 129.80, 129.30, 128.95, 127.94, 127.91, 127.88, 126.60, 115.99, 106.20, 21.12; **HRMS** (ESI) calcd for C_21_H_17_N_2_OSe [M+H]^+^: 393.0501, found: 393.0494.

***3-((4-Methoxyphenyl)selanyl)-2-phenyl-4H-pyrido[1,2-a]pyrimidin-4-one* (3p)**. 2-Phenyl-8,9-dihydro-4*H*-pyrido[1,2-*a*]pyrimidin-4-one (0.20 mmol, 44.42 mg) was reacted with 1,2-bis(4-methoxyphenyl)diselane (0.20 mmol, 74.44 mg) according to General Procedure. The crude product was purified by column chromatography (petroleum ether: ethyl acetate = 5:1) to afford the title compound as a yellow solid (m. p. 191–192 °C) in 68% yield (55.56 mg). **R*_f_*** (petroleum ether/ethyl acetate = 5:2): 0.17; **^1^H NMR** (500 MHz, CDCl_3_) *δ* 9.06 (d, *J* = 6.8 Hz, 1H), 7.77–7.74 (m, 1H), 7.70 (d, *J* = 8.6 Hz, 1H), 7.60–7.56 (m, 2H), 7.45–7.42 (m, 3H), 7.30 (d, *J* = 8.8 Hz, 2H), 7.17–7.13 (m, 1H), 6.69 (d, *J* = 8.8 Hz, 2H), 3.73 (s, 3H); **^13^C NMR** (125 MHz, CDCl_3_) *δ* 167.39, 159.11, 157.77, 149.99, 140.35, 136.55, 134.44, 129.24, 128.94, 127.90, 126.59, 121.51, 115.90, 114.60, 107.17, 55.22; **HRMS** (ESI) calcd for C_21_H_17_N_2_O_2_Se [M+H]^+^: 409.0450, found: 409.0446.

***3-((4-Chlorophenyl)selanyl)-2-phenyl-4H-pyrido[1,2-a]pyrimidin-4-one* (3q)**. 2-Phenyl-8,9-dihydro-4*H*-pyrido[1,2-*a*]pyrimidin-4-one (0.20 mmol, 44.42 mg) was reacted with 1,2-bis(4-chlorophenyl)diselane (0.20 mmol, 76.21 mg) according to General Procedure. The crude product was purified by column chromatography (petroleum ether: ethyl acetate = 5:1) to afford the title compound as a yellow solid (m. p. 187–188 °C) in 97% yield (79.88 mg). **R*_f_*** (petroleum ether/ethyl acetate = 5:2): 0.32; **^1^H NMR** (500 MHz, CDCl_3_) *δ* 9.07 (d, *J* = 7.1 Hz, 1H), 7.83–7.79 (m, 1H), 7.74 (d, *J* = 8.8 Hz, 1H), 7.60–7.55 (m, 2H), 7.45–7.40 (m, 3H), 7.25–7.18 (m, 3H), 7.11 (d, *J* = 8.5 Hz, 2H); **^13^C NMR** (126 MHz, CDCl_3_) *δ* 168.05, 157.64, 150.28, 140.08, 137.02, 132.85, 132.60, 129.92, 129.46, 129.10, 128.83, 127.96, 126.69, 116.22, 105.47; **HRMS** (ESI) calcd for C_20_H_14_ClN_2_OSe [M+H]^+^: 412.9954, found: 412.9949.

***3-((4-Bromophenyl)selanyl)-2-phenyl-4H-pyrido[1,2-a]pyrimidin-4-one* (3r)**. 2-Phenyl-8,9-dihydro-4*H*-pyrido[1,2-*a*]pyrimidin-4-one (0.20 mmol, 44.42 mg) was reacted with 1,2-bis(4-bromophenyl)diselane (0.20 mmol, 93.99 mg) according to General Procedure. The crude product was purified by column chromatography (petroleum ether: ethyl acetate = 5:1) to afford the title compound as a yellow solid (m. p. 198–199 °C) in 77% yield (70.53 mg). **R*_f_*** (petroleum ether/ethyl acetate = 5:2): 0.21; **^1^H NMR** (500 MHz, CDCl_3_) *δ* 9.07 (d, *J* = 6.7 Hz, 1H), 7.83–7.80 (m, 1H), 7.74 (d, *J* = 8.7 Hz, 1H), 7.59–7.55 (m, 2H), 7.44–7.40 (m, 3H), 7.26–7.23 (m, 2H), 7.22–7.19 (m, 1H), 7.18–7.14 (m, 2H); **^13^C NMR** (126 MHz, CDCl_3_) *δ* 168.11, 157.63, 150.30, 140.06, 137.08, 132.76, 132.01, 130.68, 129.49, 128.83, 127.97, 126.69, 120.86, 116.26, 105.28. ^19^F NMR (471 MHz, CDCl_3_) *δ* −40.57; **HRMS** (ESI) calcd for C_20_H_14_BrN_2_OSe [M+H]^+^: 456.9449, found: 456.9440.

***3-((3,5-Dimethylphenyl)selanyl)-2-phenyl-4H-pyrido[1,2-a]pyrimidin-4-one* (3s)**. 2-Phenyl-8,9-dihydro-4*H*-pyrido[1,2-*a*]pyrimidin-4-one (0.20 mmol, 44.42 mg) was reacted with 1,2-bis(3,5-dimethylphenyl)diselane (0.20 mmol, 73.65 mg) according to General Procedure. The crude product was purified by column chromatography (petroleum ether: ethyl acetate = 5:1) to afford the title compound as a yellow solid (m. p. 105–106 °C) in 95% yield (76.78 mg). **R*_f_*** (petroleum ether/ethyl acetate = 5:2): 0.32; **^1^H NMR** (500 MHz, CDCl_3_) *δ* 9.08 (d, *J* = 7.2 Hz, 1H), 7.79–7.75 (m, 1H), 7.73 (d, *J* = 8.8 Hz, 1H), 7.60 (dd, *J* = 6.5, 2.9 Hz, 2H), 7.43–7.38 (m, 3H), 7.19–7.15 (m, 1H), 6.90 (s, 2H), 6.76 (s, 1H), 2.18 (s, 6H); **^13^C NMR** (125 MHz, CDCl_3_) *δ* 167.83, 157.82, 150.17, 140.26, 138.43, 136.72, 131.24, 129.20, 128.92, 128.87, 128.74, 128.01, 127.83, 126.62, 115.99, 106.03, 21.21; **HRMS** (ESI) calcd for C_22_H_19_N_2_OSe [M+H]^+^: 407.0657, found: 407.0652.

***3-(Mesitylselanyl)-2-phenyl-4H-pyrido[1,2-a]pyrimidin-4-one* (3t)**. 2-Phenyl-8,9-dihydro-4*H*-pyrido[1,2-*a*]pyrimidin-4-one (0.20 mmol, 44.42 mg) was reacted with 1,2-dimesityldiselane (0.20 mmol, 82.60 mg) according to General Procedure. The crude product was purified by column chromatography (petroleum ether: ethyl acetate = 5:1) to afford the title compound as a yellow solid (m. p. 251–252 °C) in 40% yield (35.23 mg). **R*_f_*** (petroleum ether/ethyl acetate = 5:2): 0.45; **^1^H NMR** (500 MHz, CDCl_3_) *δ* 8.97 (d, *J* = 7.2 Hz, 1H), 7.70–7.64 (m, 2H), 7.55–7.51 (m, 2H), 7.44–7.40 (m, 3H), 7.10–7.06 (m, 1H), 6.76 (s, 2H), 2.29 (s, 6H), 2.18 (s, 3H); **^13^C NMR** (125 MHz, CDCl_3_) *δ* 165.71, 156.74, 149.12, 142.29, 140.14, 137.78, 135.70, 129.23, 128.54, 128.45, 128.13, 127.81, 127.56, 126.44, 115.49, 108.00, 24.10, 20.92; **HRMS** (ESI) calcd for C_23_H_21_N_2_OSe [M+H]^+^: 421.0814, found: 421.0808.

***3-(Naphthalen-1-ylselanyl)-2-phenyl-4H-pyrido[1,2-a]pyrimidin-4-one* (3u)**. 2-Phenyl-8,9-dihydro-4*H*-pyrido[1,2-*a*]pyrimidin-4-one (0.20 mmol, 44.42 mg) was reacted with 1,2-di(naphthalen-2-yl)diselane (0.20 mmol, 82.46 mg) according to General Procedure. The crude product was purified by column chromatography (petroleum ether: ethyl acetate = 5:1) to afford the title compound as a yellow solid (m. p. 161–162 °C) in 48% yield (41.28 mg). **R*_f_*** (petroleum ether/ethyl acetate = 5:2): 0.24; **^1^H NMR** (500 MHz, CDCl_3_) *δ* 9.05 (d, *J* = 7.1 Hz, 1H), 8.04 (d, *J* = 7.6 Hz, 1H), 7.77–7.73 (m, 2H), 7.68 (dd, *J* = 13.1, 8.5 Hz, 2H), 7.57 (dd, *J* = 7.5, 1.8 Hz, 2H), 7.53 (d, *J* = 6.5 Hz, 1H), 7.43–7.39 (m, 2H), 7.36 (q, *J* = 5.3 Hz, 3H), 7.22 (t, *J* = 7.7 Hz, 1H), 7.17–7.13 (m, 1H); **^13^C NMR** (125 MHz, CDCl_3_) *δ* 168.00, 157.75, 150.13, 140.10, 136.74, 133.97, 133.49, 131.12, 130.40, 129.28, 128.87, 128.41, 127.99, 127.93, 127.85, 127.27, 126.60, 126.29, 125.94, 125.74, 116.00, 105.86; **HRMS** (ESI) calcd for C_24_H_17_N_2_OSe [M+H]^+^: 429.0501, found: 429.0494.

***2-P**henyl-3-(pyridin-2-ylselanyl)-4H-pyrido[1,2-a]pyrimidin-4-one* (3v)**. 2-Phenyl-8,9-dihydro-4*H*-pyrido[1,2-*a*]pyrimidin-4-one (0.20 mmol, 44.42 mg) was reacted with 1,2-di(pyridin-2-yl)diselane (0.20 mmol, 62.83 mg) according to General Procedure. The crude product was purified by column chromatography (petroleum ether: ethyl acetate = 5:3) to afford the title compound as a yellow solid (m. p. 178–179 °C) in 97% yield (73.24 mg). **R*_f_*** (petroleum ether/ethyl acetate = 5:3): 0.1; **^1^H NMR** (400 MHz, CDCl_3_) *δ* 9.07 (d, *J* = 7.1 Hz, 1H), 8.32 (dd, *J* = 4.7, 0.9 Hz, 1H), 7.84–7.79 (m, 1H), 7.75 (d, *J* = 8.8 Hz, 1H), 7.64 (dd, *J* = 6.5, 3.0 Hz, 2H), 7.40–7.35 (m, 4H), 7.20 (td, *J* = 7.1, 1.3 Hz, 1H), 7.15 (d, *J* = 8.0 Hz, 1H), 7.00–6.95 (m, 1H); **^13^C NMR** (100 MHz, CDCl_3_) *δ* 168.36, 157.74, 156.65, 150.51, 150.02, 140.18, 137.20, 136.36, 129.43, 128.89, 128.02, 127.88, 126.70, 124.32, 120.59, 116.27, 104.21; **HRMS** (ESI) calcd for C_19_H_14_N_3_OSe [M+H]^+^: 380.0297, found: 380.0290.

***3-(Methylselanyl)-2-phenyl-4H-pyrido[1,2-a]pyrimidin-4-one* (3w)**. 2-Phenyl-8,9-dihydro-4*H*-pyrido[1,2-*a*]pyrimidin-4-one (0.20 mmol, 44.42 mg) was reacted with 1,2-dimethyldiselane (0.24 mmol, 37.60 mg) according to General Procedure. The crude product was purified by column chromatography (petroleum ether: ethyl acetate = 5:1) to afford the title compound as a yellow solid (m. p. 117–118 °C) in 95% yield (59.96 mg). **R*_f_*** (petroleum ether/ethyl acetate = 5:2): 0.28; **^1^H NMR** (500 MHz, CDCl_3_) *δ* 9.10–9.05 (m, 1H), 7.75–7.68 (m, 2H), 7.67–7.61 (m, 2H), 7.50–7.45 (m, 3H), 7.17 (s, 1H), 2.21 (s, 3H); ^**13**^**C NMR** (126 MHz, CDCl_3_) *δ* 165.88, 157.36, 149.36, 140.26, 135.93, 129.47, 128.93, 128.03, 127.23, 126.59, 115.85, 106.24, 7.97; **HRMS** (ESI) calcd for C_15_H_13_N_2_OSe [M+H]^+^: 317.0188, found: 317.0182.

***6-Methyl-3-(methylselanyl)-2-phenyl-4H-pyrido[1,2-a]pyrimidin-4-one* (3x)**. 6-Methyl-2-phenyl-4*H*-pyrido[1,2-*a*]pyrimidin-4-one (0.20 mmol, 47.25 mg) was reacted with 1,2-dimethyldiselane (0.20 mmol, 37.60 mg) according to General Procedure. The crude product was purified by column chromatography (petroleum ether: ethyl acetate = 5:1) to afford the title compound as yellow liquid in 94% yield (61.77 mg). **R*_f_*** (petroleum ether/ethyl acetate = 5:2): 0.26; **^1^H NMR** (500 MHz, CDCl_3_) *δ* 7.66 (dd, *J* = 7.5, 1.8 Hz, 2H), 7.44 (dd, *J* = 11.8, 5.3 Hz, 5H), 6.70 (t, *J* = 4.0 Hz, 1H), 3.06 (s, 3H), 2.12 (s, 3H); **^13^C NMR** (125 MHz, CDCl_3_) *δ* 164.41, 161.15, 151.80, 143.42, 139.93, 135.00, 129.34, 128.93, 127.97, 125.37, 118.59, 108.02, 24.65, 7.93; **HRMS** (ESI) calcd for C_16_H_15_N_2_OSe [M+H]^+^: 331.0344, found: 331.0337.

***6-(Methylselanyl)-7-phenyl-5H-thiazolo[3,2-a]pyrimidin-5-one* (3y)**. 7-Phenyl-5*H*-thiazolo[3,2-*a*]pyrimidin-5-one (0.20 mmol, 45.65 mg) was reacted with 1,2-dimethyldiselane (0.20 mmol, 37.60 mg) according to General Procedure. The crude product was purified by column chromatography (petroleum ether: ethyl acetate = 5:1) to afford the title compound as an orange solid (m. p. 147–148 °C) in 87% yield (56.03 mg). **R*_f_*** (petroleum ether/ethyl acetate = 5:2): 0.26; **^1^H NMR** (400 MHz, CDCl_3_) *δ* 8.01 (d, *J* = 4.9 Hz, 1H), 7.60 (dd, *J* = 6.5, 2.9 Hz, 2H), 7.48–7.43 (m, 3H), 7.02 (d, *J* = 4.9 Hz, 1H), 2.19 (s, 3H).; **^13^C NMR** (100 MHz, CDCl_3_) *δ* 164.71, 160.91, 158.00, 139.57, 129.59, 129.01, 127.96, 122.13, 112.15, 107.18, 8.00; **HRMS** (ESI) calcd for C_13_H_11_N_2_OSSe [M+H]^+^: 322.9752, found: 322.9746.

***3-((3-Bromophenyl)selanyl)-2-methyl-4H-pyrido[1,2-a]pyrimidin-4-one* (3z)**. 2-Methyl-4*H*-pyrido[1,2-*a*]pyrimidin-4-one (0.20 mmol, 32.04 mg) was reacted with 1,2-bis(3-bromophenyl)diselane (0.20 mmol, 93.99 mg) according to General Procedure. The crude product was purified by column chromatography (petroleum ether: ethyl acetate = 20:1) to afford the title compound as a yellow solid (m. p. 97–98 °C) in 77% yield (60.93 mg). **R*_f_*** (petroleum ether/ethyl acetate = 10:1): 0.42; **^1^H NMR** (500 MHz, CDCl_3_) *δ* 9.07–9.01 (m, 1H), 7.81–7.78 (m, 1H), 7.63 (d, *J* = 8.9 Hz, 1H), 7.46 (t, *J* = 1.7 Hz, 1H), 7.31–7.26 (m, 2H), 7.17 (td, *J* = 7.0, 1.2 Hz, 1H), 7.05 (t, *J* = 7.9 Hz, 1H), 2.73 (s, 3H).; **^13^C NMR** (126 MHz, CDCl_3_) *δ* 169.73, 157.11, 150.44, 137.31, 133.62, 132.52, 130.47, 129.63, 128.78, 128.25, 125.94, 123.13, 115.97, 104.80, 26.81; **HRMS** (ESI) calcd for C_15_H_12_BrN_2_OSe [M+H]^+^:394.9293, found: 394.9284.

***3-((3-Methoxyphenyl)selanyl)-2-methyl-4H-pyrido[1,2-a]pyrimidin-4-one* (3aa)**. 2-**M**ethyl-4*H*-pyrido[1,2-*a*]pyrimidin-4-one (0.20 mmol, 32.04 mg) was reacted with 1,2-bis(3-methoxyphenyl)diselane (0.20 mmol, 74.44 mg) according to General Procedure. The crude product was purified by column chromatography (petroleum ether: ethyl acetate = 5:2) to afford the title compound as yellow liquid in 84% yield (58.14 mg). **R*_f_*** (petroleum ether/ethyl acetate = 5:2): 0.11; **^1^H NMR** (500 MHz, CDCl_3_) *δ* 9.05 (dd, *J* = 7.1, 0.6 Hz, 1H), 7.80–7.77 (m, 1H), 7.63 (d, *J* = 8.9 Hz, 1H), 7.16 (td, *J* = 7.0, 1.2 Hz, 1H), 7.11 (t, *J* = 7.9 Hz, 1H), 6.96–6.90 (m, 2H), 6.73–6.68 (m, 1H), 3.73 (s, 3H), 2.73 (s, 3H); **^13^C NMR** (125 MHz, CDCl_3_) *δ* 169.52, 159.96, 157.19, 150.23, 137.17, 132.57, 129.93, 128.27, 125.77, 122.58, 116.03, 115.87, 112.02, 105.44, 55.27, 26.75; **HRMS** (ESI) calcd for C_16_H_15_N_2_O_2_Se [M+H]^+^: 347.0293, found: 347.0287.

***2-**Methyl-3-(p-tolylselanyl)-4H-pyrido[1,2-a]pyrimidin-4-one* (3ab)**. 2-**M**ethyl-4*H*-pyrido[1,2-*a*]pyrimidin-4-one (0.20 mmol, 32.04 mg) was reacted with 1,2-di-p-tolyldiselane (0.20 mmol, 68.04 mg) according to General Procedure. The crude product was purified by column chromatography (petroleum ether: ethyl acetate = 5:1) to afford the title compound as a yellow solid (m. p. 128–129 °C) in 73% yield (48.21 mg). **R*_f_*** (petroleum ether/ethyl acetate = 5:2): 0.15; **^1^H NMR** (500 MHz, CDCl_3_) *δ* 9.08–9.03 (m, 1H), 7.8.-7.76 (m, 1H), 7.62 (d, *J* = 8.8 Hz, 1H), 7.32 (d, *J* = 8.1 Hz, 2H), 7.16 (td, *J* = 7.1, 1.2 Hz, 1H), 7.02 (d, *J* = 7.9 Hz, 2H), 2.74 (s, 3H), 2.27 (s, 3H); **^13^C NMR** (125 MHz, CDCl_3_) *δ* 168.83, 157.14, 149.98, 137.11, 136.78, 131.26, 130.03, 128.24, 127.43, 125.63, 115.87, 106.30, 26.70, 21.07; **HRMS** (ESI) calcd for C_16_H_15_N_2_OSe [M+H]^+^: 331.0344, found: 331.0337.

***3-((4-Chlorophenyl)selanyl)-2-methyl-4H-pyrido[1,2-a]pyrimidin-4-one* (3ac)**. 2-Methyl-4*H*-pyrido[1,2-*a*]pyrimidin-4-one (0.20 mmol, 32.04 mg) was reacted with 1,2-bis(4-chlorophenyl)diselane (0.20 mmol, 76.21 mg) according to General Procedure. The crude product was purified by column chromatography (petroleum ether: ethyl acetate = 20:1) to afford the title compound as a white solid (m. p. 121–122 °C) in 67% yield (47.07 mg). **R*_f_*** (petroleum ether/ethyl acetate = 10:1): 0.51; **^1^H NMR** (500 MHz, CDCl_3_) *δ* 9.04 (d, *J* = 7.1 Hz, 1H), 7.81–7.78 (m, 1H), 7.62 (d, *J* = 8.9 Hz, 1H), 7.35–7.30 (m, 2H), 7.17 (dd, *J* = 9.1, 4.8 Hz, 3H), 2.74 (s, 3H); **^13^C NMR** (125 MHz, CDCl_3_) *δ* 169.42, 157.10, 150.31, 137.17, 132.73, 131.98, 129.66, 129.29, 128.15, 125.90, 115.90, 105.36, 26.79; **HRMS** (ESI) calcd for C_15_H_12_ClN_2_OSe [M+H]^+^: 350.9798, found: 350.9790.

***2-Methyl-3-(methylselanyl)-4H-pyrido[1,2-a]pyrimidin-4-one* (3ad)**. 2-**M**ethyl-4*H*-pyrido[1,2-*a*]pyrimidin-4-one (0.20 mmol, 32.04 mg) was reacted with 1,2-dimethyldiselane (0.20 mmol, 37.60 mg) according to General Procedure. The crude product was purified by column chromatography (petroleum ether: ethyl acetate = 5:1) to afford the title compound as a white solid (m. p. 73–74 °C) in 73% yield (37.12 mg). **R*_f_*** (petroleum ether/ethyl acetate = 5:2): 0.17; **^1^H NMR** (500 MHz, CDCl_3_) *δ* 8.99 (d, *J* = 7.1 Hz, 1H), 7.72–7.68 (m, 1H), 7.55 (d, *J* = 8.9 Hz, 1H), 7.11 (t, *J* = 6.9 Hz, 1H), 2.73 (s, 3H), 2.32 (s, 3H); **^13^C NMR** (125 MHz, CDCl_3_) *δ* 167.24, 156.77, 149.48, 136.09, 127.41, 125.79, 115.47, 106.25, 26.62, 7.16; **HRMS** (ESI) calcd for C_10_H_11_N_2_OSe [M+H]^+^: 255.0031, found: 255.0027.

***7-Methyl-6-(methylselanyl)-5H-thiazolo[3,2-a]pyrimidin-5-one* (3ae)**. 7-**M**ethyl-5*H*-thiazolo[3,2-*a*]pyrimidin-5-one (0.20 mmol, 33.24 mg) was reacted with 1,2-dimethyldiselane (0.20 mmol, 37.60 mg) according to General Procedure. The crude product was purified by column chromatography (petroleum ether: ethyl acetate = 5:1) to afford the title compound as a yellow solid (m. p. 131–132 °C) in 85% yield (43.86 mg). **R*_f_*** (petroleum ether/ethyl acetate = 5:2): 0.24; **^1^H NMR** (400 MHz, CDCl_3_) *δ* 7.92 (d, *J* = 4.9 Hz, 1H), 6.96 (d, *J* = 4.9 Hz, 1H), 2.65 (s, 3H), 2.27 (s, 3H); **^13^C NMR** (100 MHz, CDCl_3_) *δ* 166.21, 161.15, 157.52, 122.30, 111.38, 107.08, 77.40, 7.17; **HRMS** (ESI) calcd for C_8_H_9_N_2_OSSe [M+H]^+^: 260.9595, found: 260.9593.

## 4. Conclusions

We have presented a practical and sustainable C3 selenylation of pyrido[1,2-*a*]pyrimidin-4-ones under electrochemically driven external oxidant-free conditions. Various structurally diverse seleno-substituted products were obtained with broad substrate scope and with good functional group compatibility in 31 examples. A preliminary mechanism study revealed a radical pathway maybe involved under this catalytic system. Further mechanistic studies and applications of this strategy to more complicated drug candidates are underway in our laboratory.

## Data Availability

The data presented in this study are available in the article and Appendix A.

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
