# Peer review of "Electro-Oxidative C3-Selenylation of Pyrido[1,2-a]pyrimidin-4-ones"

_molecules, 2023, doi:10.3390/molecules28052206_

Round 1

Reviewer 1 Report

This manuscript reports the electrochemical  C(sp2)−H selenylation of pyrido[1,2-a]py-rimidin-4-ones by employing aryl/alkyl diselenides as the coupling substrates. The electrochemical method provides a complementary option for such C-H bond transformation to the previously known works with chemical oxidation with improved sustainability. The present method has also displayed broad application scope by the smooth synthesis of 31 selenylated products. Publication of the work is recommended after some minor revision as noted below.

1. As analogous coupling substrate of diselenide, the reaction of disulfide is suggested to see if equivalent C-H sulfenylation is applicable with the present method.

2. To make the reactions clear, it is better to indicate in the title the C-H bond site as “C3-selenylation” instead of “C(sp2)-selenylation”.

3. In the experimental section, the first main letter in the names of all products should be capitalized.

4. A few recent works reporting the TM-free C-H bond functionalization in similar substrates should be cited: Green Chem. 2022, 24, 5058-5063; Tetrahedron Lett. 2020, 61, 152226; Chin. Chem. Lett. 2021, 32, 3514-3517 etc.

Author Response

This manuscript reports the electrochemical C(sp2)−H selenylation of pyrido[1,2-a]py-rimidin-4-ones by employing aryl/alkyl diselenides as the coupling substrates. The electrochemical method provides a complementary option for such C-H bond transformation to the previously known works with chemical oxidation with improved sustainability. The present method has also displayed broad application scope by the smooth synthesis of 31 selenylated products. Publication of the work is recommended after some minor revision as noted below. 

1. As analogous coupling substrate of diselenide, the reaction of disulfide is suggested to see if equivalent C-H sulfenylation is applicable with the present method.

Answer: Thank you very much for your constructive comments. During we evaluated the scope of various diselenides, we have also attempted the C3 sulfuration of pyrido[1,2-a]pyrimidin-4-ones with disulfides. However, possibly due to the strong oxidation environment, disulfide substrates did not give the expected C3 sulfuration products.

2. To make the reactions clear, it is better to indicate in the title the C-H bond site as “C3-selenylation” instead of “C(sp2)-selenylation”.

Answer: Thank you very much for your patient corrections. Appropriate modifications have been made in the revised manuscript and supporting information.

3. In the experimental section, the first main letter in the names of all products should be capitalized.

Answer: Thank you very much for your patient corrections. We have made appropriate modification in the experimental section.

4. A few recent works reporting the TM-free C-H bond functionalization in similar substrates should be cited: Green Chem. 2022, 24, 5058-5063; Tetrahedron Lett. 2020, 61, 152226; Chin. Chem. Lett. 2021, 32, 3514-3517 etc.

Answer: Thank you very much for your constructive comments. The related works have been added in the revised manuscript.

Reviewer 2 Report

Owing to the importance of organic selenium both in biology and in organic synthesis, the development of convenient strategies for the synthesis of selenium-containing compounds is interesting. In this manuscript, an electrochemically driven approach towards seleno-containing N-heterocycles is described. The authors speculated that the success of this reaction was due to the two consecutive single-electron oxidations of the substrate through radical trapping and CV experiments. Findings reported are novel and interesting for an audience of synthetic and medicinal chemists. The substrate scope is broad and the products characterization is accurate. On the basis of these considerations, I believe that this material is suitable for publication in Molecules. 

Further comments:

1. Whether disulfides are feasible in this reaction? The authors should try it to increase the utility of the strategy.

2. In the part of substrate scope, the yield of 3t and 3u are poor. The results should be mentioned in the manuscript and the reason should be briefly commented.

3. Some recent references about chalcogen chemistry are suggested to be cited, such as Molecules2022, 27, 619; Org. Chem. Front.20229, 4536; Org. Lett., 2021, 23, 3604.

Author Response

Owing to the importance of organic selenium both in biology and in organic synthesis, the development of convenient strategies for the synthesis of selenium-containing compounds is interesting. In this manuscript, an electrochemically driven approach towards seleno-containing N-heterocycles is described. The authors speculated that the success of this reaction was due to the two consecutive single-electron oxidations of the substrate through radical trapping and CV experiments. Findings reported are novel and interesting for an audience of synthetic and medicinal chemists. The substrate scope is broad and the products characterization is accurate. On the basis of these considerations, I believe that this material is suitable for publication in Molecules.  

Further comments: 

1. Whether disulfides are feasible in this reaction? The authors should try it to increase the utility of the strategy.

Answer: Thank you very much for your constructive comments. During we evaluated the scope of various diselenides, we have also attempted the C3 sulfuration of pyrido[1,2-a]pyrimidin-4-ones with disulfides. However, possibly due to the strong oxidation environment, disulfide substrates did not give the expected C3 sulfuration products.

2. In the part of substrate scope, the yield of 3t and 3u are poor. The results should be mentioned in the manuscript and the reason should be briefly commented.

Answer: Thank you very much for your constructive comments. Appropriate modifications have been added in the revised manuscript.

3. Some recent references about chalcogen chemistry are suggested to be cited, such as Molecules, 2022, 27, 619; Org. Chem. Front., 2022, 9, 4536; Org. Lett., 2021, 23, 3604.

Answer: Thank you very much for your constructive comments. The related works have been added in the revised manuscript.

Reviewer 3 Report

The authors report the electrochemical C3-selenylation of pyrido[1,2-a]pyrimidin-4-ones under oxidant-free conditions, thus furnishing seleno-substituted N-heterocycles in good yields. This study is based on previous work from Das and co-workers, which has been cited in the manuscript. In my opinion, the synthetic novelty is quite limited, but the use of electrochemistry instead of stoichiometric oxidants is appealing and worth acceptance in Molecules. The study is clear and easy to follow, thus I would suggest only minor corrections:
1. The authors suggest a mechanism with two consecutive single-electron oxidations of the substrate, which eventually generates an extremely reactive carbocation as the key intermediate with elevated reactivity. This mechanism is different from what was reported by Das (selenium radical addition to 1a, followed by oxidation, given a carbocation more stable than the one proposed here). It would be better if the authors add more comments about it in the manuscript, trying the explain the disparity with Das’s work.
2. The supporting information seems fine, although I noticed that compound 3m doesn’t match its NMR, despite the mass being accurate. Probably the -OMe group is a -Me group: please fix it.

Author Response

The authors report the electrochemical C3-selenylation of pyrido[1,2-a]pyrimidin-4-ones under oxidant-free conditions, thus furnishing seleno-substituted N-heterocycles in good yields. This study is based on previous work from Das and co-workers, which has been cited in the manuscript. In my opinion, the synthetic novelty is quite limited, but the use of electrochemistry instead of stoichiometric oxidants is appealing and worth acceptance in Molecules. The study is clear and easy to follow, thus I would suggest only minor corrections.

1. The authors suggest a mechanism with two consecutive single-electron oxidations of the substrate, which eventually generates an extremely reactive carbocation as the key intermediate with elevated reactivity. This mechanism is different from what was reported by Das (selenium radical addition to 1a, followed by oxidation, given a carbocation more stable than the one proposed here). It would be better if the authors add more comments about it in the manuscript, trying the explain the disparity with Das’s work.

Answer: Thank you very much for your constructive comments. Appropriate modifications have been added in the revised manuscript.

It's worth noting that this mechanism is different from what was reported by Das (selenium radical addition to 1a, followed by oxidation, given a carbocation intermediate). Because the measured oxidation peak of 1a presented in 1.95V and 4.54 V, and the operating constant voltage is above 5V, so the proposed mechanism in Scheme 3 is consistent with the mechanism verification (Scheme 2 and Figure 2). Although diselenes 2 have a low oxidation potential and are easily oxidized, but they are also easily reduced at the cathode to regenerated the nucleophiles.

2. The supporting information seems fine, although I noticed that compound 3m doesn’t match its NMR, despite the mass being accurate. Probably the -OMe group is a -Me group: please fix it.

Answer:Thanks to the reviewer's patient correction, we have made appropriate modifications in the revised manuscript.